# Synthesis of Halopyrazole Matrine Derivatives and Their Insecticidal and Fungicidal Activities

**DOI:** 10.3390/molecules27154974

**Published:** 2022-08-05

**Authors:** Xingan Cheng, Huiqing He, Fangyun Dong, Chunbao Charles Xu, Hanhui Zhang, Zhanmei Liu, Xiaojing Lv, Yuehua Wu, Xuhong Jiang, Xiangjing Qin

**Affiliations:** 1Institute of Natural Product Chemistry, College of Chemistry and Chemical Engineering/Key Laboratory of Green Prevention and Control on Fruits and Vegetables in South China, Ministry of Agriculture and Rural Affairs/Institute of Plant Health, Zhongkai University of Agriculture and Engineering, Guangzhou 510225, China; 2CAS Key Laboratory of Tropical Marine Bio-Resources and Ecology, Guangdong Key Laboratory of Marine Materia Medica, South China Sea Institute of Oceanology, Chinese Academy of Sciences (CAS), Guangzhou 510301, China; 3Department of Chemical and Biochemical Engineering, Western University, London, ON N6A 5B9, Canada; 4Guangzhou Inspecting Testing and Certification Group Co., Ltd., Guangzhou 511447, China

**Keywords:** structural modification, matrine derivatives, insecticidal activity, fungicidal activity

## Abstract

Matrine is a traditional botanical pesticide with a broad-spectrum biological activity that is widely applied in agriculture. Halopyrazole groups are successfully introduced to the C13 of matrine to synthesize eight new derivatives with a yield of 78–87%. The insecticidal activity results show that the introduction of halopyrazole groups can significantly improve the insecticidal activity of matrine on *Plutella xylostella*, *Mythimna separata* and *Spodoptera frugiperda* with a corrected mortality rate of 100%, which is 25–65% higher than matrine. The fungicidal activity results indicate that derivatives have a high inhibitory effect on *Ceratobasidium cornigerum*, *Cibberella sanbinetti*, *Gibberrlla zeae* and *Collectot tichum gloeosporioides*. Thereinto, 4-Cl-Pyr-Mat has the best result, with an inhibition rate of 23–33% higher than that of matrine. Therefore, the introduction of halogenated pyrazole groups can improve the agricultural activity of matrine.

## 1. Introduction

Plant pests and diseases not only endanger food security and reduce food quality but also affect farmers’ income [1]. The promotion of organic pesticide use has significantly increased grain output, but it has also caused a series of problems such as environmental pollution, food safety and drug resistance [2]. The development of new green pesticides with high efficiency, low toxicity and easy degradation has become the current development trend.

Plant pesticides have the characteristics of diverse biological activities, low residues, low toxicity and low resistance, which are currently the main research objects of pesticide research and development [3]. However, it has shortcomings such as slow effect and short-lasting effect, so it is often used in the form of a compounding agent. In recent years, structural modification of plant-derived pesticides has become an important method for the development of green and efficient pesticides. For example, pyrethrin was a pyrethroid pesticide synthesized by the lead compound, such as allethrin, permethrin, fenvalerate and other high-efficiency biomimetic pesticides [4,5,6]. Neonicotinoid analogues are created with nicotine as the lead compound, such as thiacloprid, imidacloprid and thiamethoxam [7,8,9].

As a common plant-derived pesticide, matrine has good insecticidal, bactericidal and acaricidal activities [10]. The mode of action used for pest management is as contact poison, ingestion or gastric poison, paralyzing nerve activity and acting on the central and peripheral nervous system [11]. Studies have shown that matrine derivatives that were modified and synthesized by introducing active groups have better biological activity [12,13]. The Michael addition reaction is one of the most commonly used methods to modify the C13 and C14 positions of matrine [14,15]. Using Lawson reagent to replace the oxygen atom on the carbonyl group of matrine with a sulfur atom and undergo a nucleophilic addition reaction with the active group at position C13, a series of thiomatrine derivatives were synthesized, which have antiliver fibrosis and antihepatitis activities [16,17]. Additionally, the introduction of nitrogen-containing and sulfur-containing derivatives synthesized at the C13 position of matrine has good insecticidal activity against *Lipaphis erysimi*, *root knot nematode* and *Spodoptera frugiperda* [18,19,20]. The derivative that was obtained by introducing dimethyl carbonate group condensed ester at the C14 position of matrine, and then reduced to alcohols and decarbonylated by lithium aluminum hydride has a significantly higher antitobacco mosaic virus and plant fungicidal activities than matrine [21]. Matrine is easy to acidolyze, and the acid derivatives have strong insecticidal and acaricidal activities, especially with the introduction of 1,3,4-thiadiazole, 4-methylbenzyl and 2-chlorobenzyl active groups [22,23]. Additionally, modified with coumarin and piperazine groups, the acid derivatives can be used to treat cancer and tumors [24,25].

Herein, we report the modification of the matrine to improve its insecticidal and fungicidal activities. Nitrogen-containing heterocyclic compounds are currently the main intermediates in the research of new pesticides, especially pyrazoles with broad-spectrum activity [26]. Currently, commonly used pyrazole insecticides and fungicides include carbamate pyrazoles, phosphate pyrazoles, pyrazole amides and oxime ether pyrazoles [27,28,29,30]. Pyrazole and its halogenated active groups are often used in pesticide intermediates [31,32]. The introduction of halogenated pyrazole groups to modify the structure can not only improve the biological activity of the lead compound but also improve its solubility, stability and reduce drug resistance [33,34]. Accordingly, halogenated pyrazole groups were introduced onto C13 of the matrine to synthesize halopyrazole derivatives. Finally, the optimization of the reaction conditions, the structural characterization, and the insecticidal and fungicidal activities were studied.

## 2. Results and Discussion

### 2.1. Chemical Synthesis

The nitrogen atom in the pyrazole group contains a lone pair of electrons, which will attack the C13 positive ions in matrine and synthesize 13-pyrazole matrine derivatives through nucleophilic addition. The experiment explored the material ratio, catalyst, temperature, solvent, etc., and found the best scheme for pyrazole derivative synthesis (Table 1). Seven pyrazole matrine derivatives were synthesized by Michael addition with a yield of 75–85%. It can be seen from Appendix A (ESI†) that the retention times of the 7 compounds were 4.141, 4.356, 4.401, 4.071, 4.346, 4.424 and 4.539 min, respectively. Compared with matrine (4.239 min) and sophocarpine (4.291 min), the peak of retention times was shifted. Therefore, it can be inferred that the obtained compounds are the target product.

### 2.2. Spectroscopic Characterization

The IR spectra of sophocarpine and compounds **1**–**7** are shown in Appendix A (ESI†). The characteristic bands of sophocarpine at 1660 cm^−1^ and 1599 cm^−1^ can be assigned to carbonyl and the carbon–carbon double bonds. Additionally, the strong peaks at about 1635 cm^−1^ were assigned to carbonyl absorption of compounds **1**–**7**. Compared with sophocarpine, the characteristic peaks of the carbonyl group are red-shifted because of the introduction of the target group. In addition, the characteristic bands at about 3100 cm^−1^ can be attributed to the ν(C-H) absorption of the carbon–carbon double bond of the pyrazolyl. Additionally, the peaks at 600–800 cm^−1^ are assigned to C-X absorption of compounds **1**–**7**, respectively. The detection of these IR peaks further confirms that the halopyrazole groups were successfully introduced into the matrine molecule.

The mass spectra of compounds **1**–**7** obtained are shown in Appendix A (ESI†). The molecular formula of compounds **1**–**7** were determined as C_18_H_25_FN_4_O, C_18_H_25_ClN_4_O, C_18_H_25_BrN_4_O, C_18_H_25_IN_4_O on the basis of ESI-MS ion peaks at m/z 333.8 ([M+H]^+^, calcd 332.20), 349.8 ([M+H]^+^, calcd 348.17), 393.6 ([M+H]^+^, calcd 392.1) and 441.4 ([M+H]^+^, calcd 440.1) respectively.

The NMR spectra of compounds **1**–**7** are shown in Appendix A (ESI†). Two triplet states at about 4.6 ppm in the ^1^H NMR spectrum can be assigned to the C13-H of compounds **1**–**7** (ESI†, Appendix A). The chemical shifts of protons in the carbon-carbon double bond of the pyrazolyl group were observed at 6.0–7.5 ppm. Additionally, in the ^13^C NMR spectrum analysis, the chemical shift peak at about 50 ppm can be attributed to the C13 from matrine. The singlet at about 165 ppm can be assigned to the carbon atom of corbonyl in compounds **1**–**7**. The characteristic peaks at 65–145 ppm in the ^13^C NMR spectra can be assigned to the pyrazolyl of compounds **1**–**7** (ESI†, Appendix A). Particularly, the characteristic peaks at 125–130 ppm were assigned to the carbon atoms of C-X bonds. It is concluded that the halopyrazole groups were formed with the C13 of matrine.

### 2.3. Structural Descriptions

The crystallographic data for the structures of Compounds **1**–**7** have been deposited with the Cambridge Crystallographic Data Centre as supplementary publication numbers CCDC 1977598, 1977599, 1977605, 1975040, 1876466, 1975042 and 1975043, respectively. Some key crystallographic data of compounds **1**–**7** are provided in Appendix A (ESI†). As shown in Table 1, the results of the empirical formula and formula weight determined by the crystallographic data are consistent with the characterization results of the MS and NMR as presented previously. As indicated in Appendix A (ESI†) halopyrazole groups were successfully introduced into the C13 of matrine. Compared with sophocarpine, the introduction of halopyrazole groups caused the nearby bond length to stretch and a bond angle to shrink. It can be seen from Appendix A that halopyrazole groups and matrine are connected by C13-N19 with a bond length of about 1.46 Å, forming the bond angle C14-C13-N19 of about 110°. The bond angles C23-N19-N20 (113°), C21-N20-N19 (103°), C23-C22-C21 (105°) and C22-C21-N20 (113°) formed by the bonds of N19-C13 (1.46 Å), N20-C21 (1.32 Å) and C22-C23 (1.37 Å) are assigned to pyrazolyl, while the halogen atom and pyrazole connected by C21-F24 (1.348 Å), C22-Cl24 (1.72 Å), C21-Br24 (1.90 Å), C22-I24 (2.08 Å). It can be observed from Appendix A (ESI†) that there is no hydrogen bonding between the molecules of the compounds, and the molecules are arranged in an orderly manner.

### 2.4. Insecticidal Activities

Figure 1 shows the corrected mortality of the compounds against common second instar lepidopteran pests. The results indicated that the introduction of halogenated pyrazole groups significantly improved the activity of matrine against *P. xylostella*, with corrected mortality rates of up to 100% for 3-Cl-Pyr-Mat, 3-Br-Pyr-Mat, 4-F-Pyr-Mat and 4-I-Pyr-Mat. The LC_50_ of the 3-halogenated pyrazole matrine derivatives was much less than 0.01 mg/L, indicating that the introduction of 3-halogenated was more effective than that of 4-halogenated (Table 2). The corrected mortality rate of matrine was as high as 96% at 1.00 mg/mL and the LC_50_ showed that the introduction of the 3-chloropyrazole group has an effect on improving the insecticidal activity of matrine against *S. exigua.* Moreover, as compared with matrine (with a corrected mortality of 62.07%), 3-Br-Pyr-Mat, 4-Cl-Pyr-Mat and 4-I-Pyr-Mat exhibited potent pesticidal activities against *S. litura*, with a corrected mortality of 76.92.3%, 88.46% and 73.08%, respectively (Figure 1). Evidently, the introduction of halopyrazole groups improved the insecticidal activity of matrine derivatives against *M. separata*. The halogenated substitution at position three was better than at position four, with a corrected mortality rate of up to 96% for the derivatives, which was 65% higher compared to matrine (31.03%). From Table 1, the LC_50_ of most of the derivatives was less than 0.01 mg/mL, which was significantly lower compared to the Matrine (more than 1.00 mg/mL). The insecticidal activity of Matrine against *S. frugiperda* was significantly increased by the introduction of 3-halogenated pyrazole groups, with corrected mortality rates of up to 100% for 3-Cl-Pyr-Mat and 3-I-Pyr-Mat. It can be concluded that the introduction of halopyrazole groups can significantly improve or enhance the insecticidal activity of matrine.

### 2.5. Fungcidal Activities

Figure 2 shows the inhibitory effect of matrine and its pyrazole derivatives on six common plant fungi. From the figure, we can see that matrine has low activity against the five plant fungi, except for *G. sanbinetti*, and that the introduction of halogenated pyrazole groups increased the fungicidal activities. The inhibition rate was reduced by 10–20% against *B. sorokiniana, C. cornigerum, G. sanbinetti, FOC and CTG* when the active group were introduced, especially 3-bromopyrazole and 4-chloropyrazole were the most effective. Moreover, as compared with matrine (with an inhibition rate of 33.96%), 4-Cl-Pyr-Mat exhibited potent activity against *G. zeae* with a corrected mortality of 67.14%. More interestingly, we found that the introduction of chloropyrazole and bromopyrazole groups were more effective. Additionally, the halogenation at C4 was better than that at C3 when the chlorinated and iodinated pyrazole groups was introduced, while the halogenation at C3 was better than that at C4 when the brominated pyrazole groups were introduced. Thus, the introduction of the halopyrazole groups increased the fungicidal activity of matrine.

### 2.6. Degradability

It can be seen that most derivatives have a higher residue rate than matrine at 60 min (ESI†, Appendix A). However, when the exposure time was up to 150 min, the residual rate of all derivatives was lower than matrine. It indicated that the derivatives synthesized by introducing halopyrazole groups were environmentally friendly.

## 3. Materials and Methods

### 3.1. Materials

Sophocarpine was purchased from Bidepharm Co., Ltd., Shanghai, China. Cesium carbonate, tripotassium orthophosphate and halodopyrazole were purchased from Aladdin Biochemical Technology Co., Ltd., Shanghai, China. Ethyl acetate and acetonitrile were obtained from Fuyu chemical Co., Ltd., Tianjin, China. Dimethyl sulfoxide (DMSO) was supplied by American Sigma-Aldrich Co., Ltd. (St. Louis, MO, USA). Ethanol, methanol, dichloromethane, etc. were purchased from Tianjin Damao Chemical Reagent Factory. Chlorpyrifos and carbendazim were purchased from Guangdong Wengjiang Chemical Reagent Co., Ltd. and Beide Pharmaceuticals, respectively. Potato Dextrose Agar (PDA) was purchased from Guangdong Huankai Microbial Technology Co., Ltd.

*Plutella xylostella (P. xylostella), Spodoptera exigua Hiibner (S. exigua), Spodoptera litura (S. litura), Mythimna separata (M. separata) and Spodoptera frugiperda (S. frugiperda)* were purchased from KE YUN Bio. *Bipolaris sorokiniana (B. sorokiniana), Ceratobasidium cornigerum (C. cornigerum), Gibberella sanbinetti (G. sanbinetti), Gibberella zeae (G. zeae), Fusarium. Oxysporum *f. sp.* Cucumerinu**, FOC (FOC)* and *Colletot tichum gloeosporioides (CTG),* which identified by sequencing were all from the South China Sea Institute of Oceanology, Chinese Academy of Sciences.

### 3.2. Synthesis of Compounds 1–7

A series of matrine derivatives were synthesized by introducing pyrazole and halogenated pyrazole groups at the C13 position of sophocarpine by the Michael addition reaction. The nitrogen atom in the pyrazole group contains a lone pair of electrons, which will attack the C13 carbocation of matrine to form a matrine derivative. Sophocarpine, catalyst and pyrazole or halogenated pyrazole were charged in a 50 mL 3-neck round-bottom flask equipped with condensation circulation device. Reaction was stirred vigorously in the solvent system and monitored via thin layer chromatography (TLC). Products were visualized by UV light (254 nm) and/or with 0.5% I_2_ stain. The target product was separated by column chromatography and purified by recrystallization.

### 3.3. Analytical Methods

HPLC chromatograms were obtained by Agilent1200 equipped with chromatograph column CNW Athena C18-WP (4.6 × 250 mm, 5 µm) and UV detector. The mobile phase consisted of ethanol/0.1% tripotassium phosphate/acetonitrile 10/10/80 (*v*/*v*) and was delivered at a flow rate of 1.0 mL/min. Infrared (IR) spectra were collected on a Spectrum 100 (Perkin Elmer) with KBr disk method, ^1^H NMR and ^13^C NMR spectra on BRUKER AVANCE III HD 700 or 176 MHz spectrometers using CDCl_3_ as the solvent, and low-resolution electrospray mass spectra (LR-ESI-MS) on BRUKER AmaZon SL. X-ray diffraction patterns of the matrine derivatives were obtained with a Smart 1000 CCD diffractometer. The chemical structures were drawn by Chemdraw, IR spectra were processed by Origin 2019 and the mass spectra by Qualitative Analysis of Mass HuterAcquisition Data. The NMR spectroscopies were analyzed by MestReNova. Single crystal diffraction data were obtained by Olex 2.

### 3.4. Insecticidal Activity

Different compounds at varying concentrations were used to test the insecticidal toxicity against agricultural pests by leaf-dipping method. The non-contaminated leaves were cut into pieces of the same dimensions and dipped into the different test solutions from 5 to 10 s before removal to dry naturally. Finally, the dried leaves and 10 pests were placed in a petri dish with filter paper and the experiment was repeated 3 times at each concentration. The petri dishes were placed in an incubator ((28 ± 1) °C, RH = 80%) and evaluated for 24, 48, 72 h. Mortality and corrected mortality were calculated according to the Abbott formula, and LC_50_ and virulence regression equations were calculated based on the probabilistic analysis.

### 3.5. Fungcidal Activity

The inhibitory effect on phytopathogenic fungi was tested by the growth rate method. With 0.5% dimethyl sulfoxide as solvent, compounds were added to sterilized PDA medium and mixed, and finally 0.25, 0.50, 0.75, 1.00 and 1.25 mg/mL drug-containing medium were prepared. Inoculated the fungi cake into the medicated medium and placed the petri dish in an incubator at 25 °C for 1 week. The size of fungi circle was recorded by the cross method.

### 3.6. Degradability Test

The derivatives that were dissolved and set to 5–6 concentration gradients were detected by high-performance liquid chromatograph to make a standard curve. The derivatives were dissolved in ethanol and illuminated by a 300 W mercury lamp with a wavelength of 365 nm. Samples were sampled and tested every half an hour and the drug concentration was calculated through the standard curve.

## 4. Conclusions

In this study, seven new matrine derivatives were synthesized by introducing pyrazole and halopyrazole groups onto C13 of matrine with a yield of 78–87%. Derivatives exhibited much stronger insecticidal and fungicidal activities than that of matrine, due to the introduction of halopyrazole groups. Additionally, the introduction of halogenated pyrazoles was more effective than pyrazoles, which concludes that the halogen atom probably acts as a key role.

## Figures and Tables

**Figure 1 molecules-27-04974-f001:**
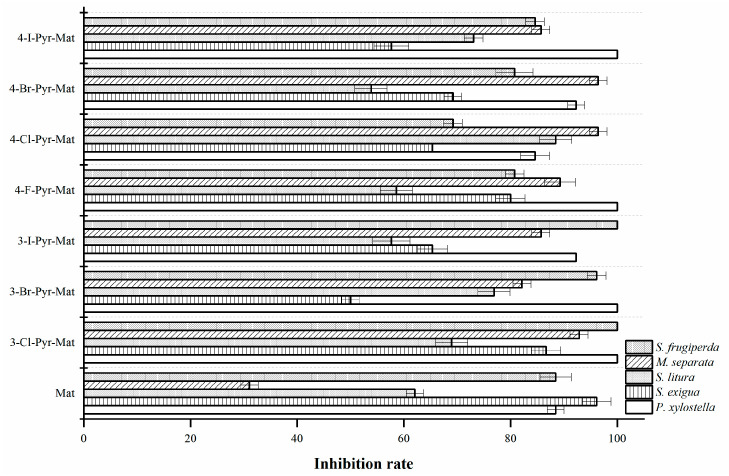
Insecticidal activity of matrine and its derivatives against agricultural pests at concentration of 1.0 mg/mL.

**Figure 2 molecules-27-04974-f002:**
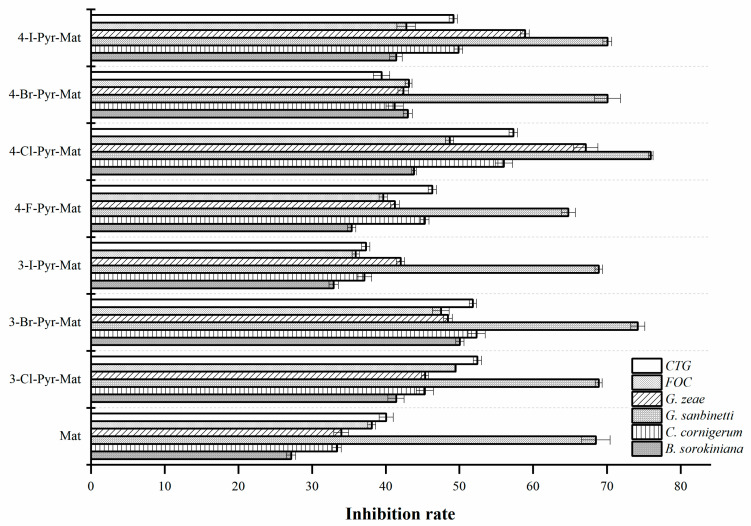
Fungicidal activities of matrine and its derivatives at concentration of 1.0 mg/mL.

**Table 1 molecules-27-04974-t001:** The reaction conditions.

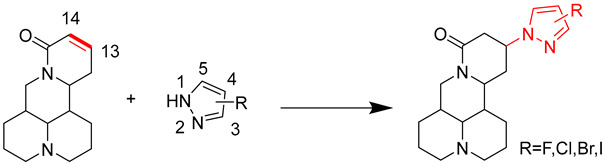					
Number	Entry	Material Ratio ^1^	Solvent	Catalyst	Yield/%
Compound 1	3-Cl-Pyr-Mat	1:1	Acetonitrile	K_3_PO_4_	82.8
Compound 2	3-Br-Pyr-Mat	1:1	Ultrapure water	Cs_2_CO_3_	84.5
Compound 3	3-I-Pyr-Mat	1:1	Acetonitrile	K_3_PO_4_	78.1
Compound 4	4-F-Pyr-Mat	2:1	Dioxane	K_3_PO_4_	85.6
Compound 5	4-Cl-Pyr-Mat	2:1	Dioxane	Cs_2_CO_3_	87.0
Compound 6	4-Br-Pyr-Mat	4:3	Dioxane	Cs_2_CO_3_	83.0
Compound 7	4-I-Pyr-Mat	1:1	Acetonitrile	Cs_2_CO_3_	81.3

^1^ The material ratio was ethyl sophocarpine/pyrazolyl or halopyrazol.

**Table 2 molecules-27-04974-t002:** Toxicities of matrine and its derivatives to agricultural pests.

	*P. xylostella*	*S. exigua*	*S. litura*	*M. separata*	*S. frugiperda*
Mat	0.052	0.043	0.215	>1.00	<0.01
3-Cl-Pyr-Mat	<0.01	<0.01	0.189	<0.01	<0.01
3-Br-Pyr-Mat	<0.01	0.75	0.198	<0.01	<0.01
3-I-Pyr-Mat	<0.01	0.652	0.444	<0.01	<0.01
4-F-Pyr-Mat	<0.01	0.058	0.316	0.012	<0.01
4-Cl-Pyr-Mat	0.035	0.385	0.095	0.020	0.083
4-Br-Pyr-Mat	0.014	0.297	>1.00	<0.01	0.032
4-I-Pyr-Mat	0.01	0.414	0.228	<0.01	0.058

## Data Availability

Not applicable.

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
