# Peer review of "Synthesis of Halopyrazole Matrine Derivatives and Their Insecticidal and Fungicidal Activities"

_molecules, 2022, doi:10.3390/molecules27154974_

Round 1
Reviewer 1 Report
This manuscript describes the full details of the syntheses of halopyrazole substituted matrine derivatives and these insecticidal and fungicial activities. According to the biological test, the new compound would be a promising candidate for the new pesticides. Chemistries described in this paper would be a nice piece of work and of interest for the researchers in this field.
I would recommend this manuscript for publication in Molecules after suitable revision listed below.
In general, citation numbers should to be shown in the text. The readers cannot find suitable references.
All of the 13-substituted matrine derivatives were prepared from the Michael addition of sophocarpine with halo-pyrazoles. Many readers would be interested in the stereoselectivity of this reaction. It should be discussed.
Author Response
Point 1: In general, citation numbers should to be shown in the text. The readers cannot find suitable references.
Response 1: We are very sorry for our negligence, and the citation numbers were added in the test.
Point 2: All of the 13-substituted matrine derivatives were prepared from the Michael addition of sophocarpine with halo-pyrazoles. Many readers would be interested in the stereoselectivity of this reaction. It should be discussed.
Response 2: Thank you for your suggestion. The catalyst used in the synthesis reaction in the article were inorganic salts, which were achiral catalysts, and the obtained products were racemate, so there is no discussion of stereoselectivity.
Reviewer 2 Report
The authors reported the synthesis of novel matrine derivatives bearing (halo)pyrazole ring at 13 position. The authors also investigated their insecticidal and fungicidal activities. Better activities were obtained from the compounds having halopyrazole ring compared with the compounds with pyrazole ring and without pyrazole ring (matrine). However, the determination of the compound bearing pyrazole ring (Pyr-Met) was doubtable. Mass peak should be observed at 314.21 for [M]+ (C18H26N4O) or at 315.22 for [M+H]+ (C18H27N4O). But the authors found the mass peak at 317.9. Furthermore, the integration of pyrazole moieties (at 7.71 and 7.52 ppm) was too small to assign two protons in 1H NMR spectrum. Same interrogation was arisen at 4.30 ppm. Thus, the correction of the data should necessitate to publish this manuscript in Molecules.
Further consideration should be required.
1) In title, “Fungicial” should be changed into “Fungicidal”.
2) The compound number (1-8) should be assigned to each compound in the text.
3) Please clarify the reference number in the main text. Furthermore, there was no reference number for “Isman, 2015” (shown in line 45), “Xu et al., 2014; Li et al., 2019” (shown in line 61), and “Cheng et al., 2018; Cheng et al., 2020; He et al., 2020” (shown in lines 67-68).
4) Please indicate the detail of the operation for HPLC (solvent and eluent speed).
5) The authors commented the smooth degradation to lead the environmentally friendly features. But the side product from the degradation would be possible to act as non-environmentally friendly influence.
6) The title in Supplementary Information did not same in the manuscript.
Author Response
Point 1: The authors reported the synthesis of novel matrine derivatives bearing (halo)pyrazole ring at 13 position. The authors also investigated their insecticidal and fungicidal activities. Better activities were obtained from the compounds having halopyrazole ring compared with the compounds with pyrazole ring and without pyrazole ring (matrine). However, the determination of the compound bearing pyrazole ring (Pyr-Mat) was doubtable. Mass peak should be observed at 314.21 for [M]+ (C18H26N4O) or at 315.22 for [M+H]+ (C18H27N4O). But the authors found the mass peak at 317.9. Furthermore, the integration of pyrazole moieties (at 7.71 and 7.52 ppm) was too small to assign two protons in 1H NMR spectrum. Same interrogation was arisen at 4.30 ppm. Thus, the correction of the data should necessitate to publish this manuscript in Molecules.
Response 1: Thanks for your professional question. We did a lot of conditional exploration in the early stage, but can not get a high purity Pyr-Mat. Based on our empirical analysis, it was inferred that the product was Pyr-Mat. First, there are active nitrogen and hydrogen in the pyrazole molecule, which is prone to tautomerism, resulting in complex reaction products and increasing the difficulty of purification. Secondly, the mainly discusses in this paper was the synthesis and activity of halogenated pyrazole matrine. Pyr-Mat was used as a comparison, so we consider adding it to the paper at last.
Point 2: In title ”Fungicial” should be changed into “Fungicidal”.
Response 2: Thanks for your suggestion, and the word have been changed.
Point 3: The compound number (1-8) should be assigned to each compound in the text.
Response 3: Considering the your suggestion, we have been added compound number to table 1.
Point 4: Please clarify the reference number in the main text. Furthermore, there was no reference number for “Isman, 2015” (shown in line 45), “Xu et al., 2014; Li et al., 2019” (shown in line 61), and “Cheng et al., 2018; Cheng et al., 2020; He et al., 2020” (shown in lines 67-68).
Response 4: We are very sorry for our negligence, and the citation numbers and references were added in the test.
Point 5: Please indicate the detail of the operation for HPLC (solvent and eluent speed).
Response 5: Thanks for your suggestion, we have been added the operation for HPLC to section 2.3 of the article.
Point 6: The authors commented the smooth degradation to lead the environmentally friendly features. But the side product from the degradation would be possible to act as non-environmentally friendly influence.
Response 6: Thank you for your innovative question, which is what we will need to focus on in the future. It is speculated from the structure that the degradation of the compound contains matrine and halogenated pyrazole groups. Matrine is a popular botanical pesticide, which is environmentally friendly. And halogenated pyrazoles are also used as intermediate compounds in the synthesis of commonly used pesticides. So it is inferred that it will not cause much impact on the environment.
Point 7: The title in Supplementary Information did not same in the manuscript.
Response 7: We have changed this part according to the Reviewer’s suggestion.
Round 2
Reviewer 1 Report
I accept the careful revision done by the authors although a misunderstanding is still remaining.
The substrate employed for the Michael addition described in this paper is sophocarpine, which is a chiral alkaloid. The Michael addition to this substrate provides an additional chiral carbon. So, the product would be a mixture diastereomers even racemic sophocarpine was used as a substrate. However, this point would not be affected on the value of the revised manuscript.
Author Response
Point 1: The substrate employed for the Michael addition described in this paper is sophocarpine, which is a chiral alkaloid. The Michael addition to this substrate provides an additional chiral carbon. So, the product would be a mixture diastereomers even racemic sophocarpine was used as a substrate. However, this point would not be affected on the value of the revised manuscript.
Response 1: Thanks for your professional question, and we agree that the product may be a mixture diastereomers. There is no chiral separation of the product, so we do not discuss it in depth.
Reviewer 2 Report
By reviewing the revised manuscript, I felt that the authors did not add the comment about low purity and unsatisfied assignment for compound 1. Parent mass should be 314.21 for [M]+ (C18H26N4O), or 315.22 in the case of [M+H]+ (molecular formula; C18H27N4O+ = 12.01 x 18 + 1.01 x 27 + 14.01 x 4 + 16.00 = 315.49). But the authors still assigned it as 316.22 for C18H27N4O+ and 317.9 for found data (lines 179-180). This is scientifically wrong. Proton NMR spectrum in SI also give unsatisfied assignment. The authors did not revise the addition of the comment of not-isolated sample for compound 1. If this manuscript was published at this version, the readers would misunderstand about the activity of those compounds. Although the main target in this manuscript is halogenated compound, the comparison between non-halogenated and halogenated compounds might be misleading about the role of halogen atom in the activity.
I strongly recommend the addition of precious information about the compound 1, or delete the discussion compared with the compound 1 to publish in Molecules.
Additional mistake:
In line 29, 76, 88, 140, 239, 243, 252, 254, 263, “fungicial” was still remained. Please revise to “fungcidal”.
Author Response
Point 1: By reviewing the revised manuscript, I felt that the authors did not add the comment about low purity and unsatisfied assignment for compound 1. Parent mass should be 314.21 for [M]+ (C18H26N4O), or 315.22 in the case of [M+H]+ (molecular formula; C18H27N4O+ = 12.01 x 18 + 1.01 x 27 + 14.01 x 4 + 16.00 = 315.49). But the authors still assigned it as 316.22 for C18H27N4O+ and 317.9 for found data (lines 179-180). This is scientifically wrong. Proton NMR spectrum in SI also give unsatisfied assignment. The authors did not revise the addition of the comment of not-isolated sample for compound 1. If this manuscript was published at this version, the readers would misunderstand about the activity of those compounds. Although the main target in this manuscript is halogenated compound, the comparison between non-halogenated and halogenated compounds might be misleading about the role of halogen atom in the activity. I strongly recommend the addition of precious information about the compound 1, or delete the discussion compared with the compound 1 to publish in Molecules.
Response 1: Thanks for your professional question. We have deleted the discussion about the compound 1 .
Point 2: Additional mistake: In line 29, 76, 88, 140, 239, 243, 252, 254, 263, “fungicial” was still remained. Please revise to “fungcidal”.
Response 2: Thanks for your suggestion, and the word have been changed.